# Comparative Effectiveness of Ampicillin/Sulbactam versus Cefazolin as Targeted Therapy for Bacteremia Caused by Beta-Lactamase-Producing Methicillin-Sensitive *Staphylococcus aureus*: A Single-Center Retrospective Study

**DOI:** 10.3390/antibiotics11111505

**Published:** 2022-10-28

**Authors:** Jun Hirai, Nobuhiro Asai, Mao Hagihara, Takaaki Kishino, Hideo Kato, Daisuke Sakanashi, Wataru Ohashi, Hiroshige Mikamo

**Affiliations:** 1Department of Clinical Infectious Diseases, Aichi Medical University Hospital, Nagakute 480-1103, Japan; 2Department of Infection Control and Prevention, Aichi Medical University Hospital, Nagakute 480-1103, Japan; 3Department of Molecular Epidemiology and Biomedical Sciences, Aichi Medical University, Nagakute 480-1103, Japan; 4Department of Emergency and Critical Care Medicine, Aichi Medical University Hospital, Nagakute 480-1103, Japan; 5Department of Pharmacy, Mie University Hospital, Tsu 514-8507, Japan; 6Division of Biostatistics, Clinical Research Center, Aichi Medical University, Nagakute 480-1103, Japan

**Keywords:** beta-lactamase-producing methicillin-sensitive *Staphylococcus aureus*, bacteremia, ampicillin/sulbactam, cefazolin, definitive therapy, Japan

## Abstract

Cefazolin (CFZ) is the first-line treatment for beta-lactamase-producing methicillin-sensitive *Staphylococcus aureus* (BP-MSSA) infection. In 2019, Japan experienced a CFZ shortage because of foreign object inclusion in a batch. Ampicillin/sulbactam (SAM) was preferred in many cases as definitive therapy for the treatment of BP-MSSA bacteremia to preserve broad-spectrum antibiotic stock. However, there are no previous studies reporting the clinical efficacy of SAM for BP-MSSA bacteremia. We aimed to compare the clinical efficacy and adverse effects of SAM versus CFZ in patients with BP-MSSA bacteremia. In total, 41 and 30 patients treated with SAM and CFZ, respectively, were identified. The baseline characteristics were similar in both groups. No significant differences were observed in length of hospital stay and all 30-day mortality between the two groups (*p* = 0.270 and 0.643, respectively). Moreover, no intergroup difference in 90-day mortality was found (hazard ratio 1.02, 95% confidential interval 0.227–4.53). Adverse effects, such as liver dysfunction, were less in the CFZ group than in the SAM group (*p* = 0.030). Therefore, in cases of poor CFZ supply or in patients allergic to CFZ and penicillinase-stable penicillins, SAM can be an effective therapeutic option for bacteremia due to BP-MSSA with attention of adverse effects, such as liver dysfunction.

## 1. Introduction

*Staphylococcus aureus* (*S. aureus*) is a Gram-positive grape-shaped bacterium that causes severe bacteremia in both community- and hospital-acquired infections [1,2]. In particular, *S. aureus* bacteremia (SAB), which sometimes complicates disseminated infections, such as pyogenic spondylitis and deep-seated abscesses, is associated with a high rate of early mortality at 10–30% [1,3]. According to the Japan Nosocomial Infections Surveillance, which is one of the largest national antimicrobial resistance surveillance systems worldwide, the clinical isolation rate of β-lactamase-producing methicillin-susceptible *S. aureus* (BP-MSSA) is more than 50% among methicillin-susceptible *S. aureus*, and this proportion is similar to that in other countries [4,5]. Thus, penicillinase-stable penicillins (PSP), such as nafcillin, oxacillin, cloxacillin, and flucloxacillin, are globally the first choice for the standard treatment of BP-MSSA [6]. However, these agents have not yet been approved in Japan. Therefore, cefazolin (CFZ), which has recently been reported to have clinical efficacy similar to that of PSP [7,8,9], is the first-line treatment for bacteremia caused by BP-MSSA.

However, in 2019, Japan encountered a CFZ shortage owing to foreign object inclusion and the outage of chemical precursors [10,11] that lasted for more than 1.5 years, following which the CFZ supply system was restored in Japan. During the time of CFZ shortage, the Ministry of Health, Labour and Welfare of Japan recommended some alternative antimicrobial agents, such as ampicillin/sulbactam (SAM), cefotaxime (CTX), ceftriaxone (CRO), and vancomycin (VAN), for SAB treatment [12]. To preserve broad-spectrum antibiotic stock, many facilities in Japan have been forced to provide targeted treatment with SAM as an alternative definitive therapy for BP-MSSA infection, although no previous studies have reported the clinical efficacy of SAM for BP-MSSA bacteremia. 

This study aimed to evaluate the clinical efficacy and adverse effects of SAM compared with those of CFZ against BP-MSSA bacteremia. The comparison of these two drugs in clinical practice could bridge a knowledge gap as shortage of antimicrobial agents can occur everywhere, and first-line treatments would be unavailable.

## 2. Results

During the study period, we enrolled 71 patients who met the inclusion criteria, of whom 41 received SAM and 30 received CFZ (Figure 1).

The demographics of the two subgroups in this study are summarized in Table 1. There was no significant intergroup difference in age, sex, or body mass index. Half of the CFZ cohort had hospital-acquired bacteremia, whereas in the SAM cohort, 16 of 41 (39%) patients had hospital-acquired BP-MSSA. The proportion of Eastern Cooperative Oncology Group (ECOG) Performance Status (PS) 3–4 was 43.9% and 30.0% in the SAM and CFZ groups, respectively. Considering the Quick Sepsis-related Organ Failure Assessment (qSOFA) score, the proportion of qSOFA ≥2 showed no intergroup difference. Although complications of septic shock occurred in approximately 10% of patients in each group, only one patient who was treated with SAM used vasopressors. Disseminated intravascular coagulation was more likely to develop in the SAM group than in the CFZ group (41.5% vs. 26.7%, *p* = 0.197). No significant difference was observed between the two groups with regard to underlying disease, including Charlson Comorbidity Index (CCI) (≥4) and laboratory data. Regarding the infection source, 13 of 41 (31.7%) patients in the SAM group and 12 of 30 (40%) patients in the CFZ group had bacteremia of unknown origin. In both groups, the most frequent origin of secondary bacteremia was catheter-associated bloodstream infection (CABSI), followed by skin and soft tissue infections. Metastatic infections occurred in 6 of 41 (14.6%) patients in the SAM group and in 3 of 30 (10%) patients in the CFZ group. Only two patients had continuous positive blood cultures in each group. On comparing both groups, the duration of intravenous antimicrobial agent administration and of total treatment did not differ significantly. The clinical outcomes and antibiotic-related adverse reactions were recorded for each patient. 

With regard to all-cause 30-day mortality, 9.8% and 6.7% of the patients in the SAM and CFZ groups, respectively, died (*p* = 0.643). Six patients were considerably older (age range, 72–97 years), and five of six patients (83.3%) had hospital-acquired BP-MSSA. The origin of infection in those six patients was CABSI (*n* = 3) or an unknown source of bacteremia (*n* = 3). Figure 2 shows the Kaplan–Meier curve of the survival rates for both the SAM and CFZ cohorts. The 90-day mortality rates were similar in the SAM and CFZ groups (12.2% vs. 13.3%, respectively, *p* = 1.0, hazard ratio 1.02, 95% confidential interval 0.227–4.53).

Table 2 shows the initial antimicrobial agents that were administered as empirical therapy. The most frequently prescribed antimicrobial agents were SAM (*n* = 14, 19.7%), followed by piperacillin/tazobactam (*n* = 12, 17.0%), CRO (*n* = 11, 15.5%), meropenem (*n* = 8, 11.3%), and VAN (*n* = 8, 11.3%). In the SAM group, 11 of 41 (26.8%) patients received SAM as both empirical and definitive therapy. However, only five patients (16.7%) received CFZ as empirical and definitive therapy. Among patients who were administered SAM as the initial antibiotic treatment (*n* = 14), three (21.4%) were switched from SAM to CFZ because of adverse effects. However, in patients who were administered CFZ as empirical therapy (*n* = 5), the CFZ treatment regimen was completed.

With regard to adverse drug reactions, the SAM cohort was more likely to experience adverse effects than the CFZ cohort (41.5% vs. 16.7%, *p* = 0.026). In particular, as shown in Table 3, liver dysfunction was more likely to complicate the clinical course in the SAM group than in the CFZ group (*p* = 0.030). None of the patients had complications of *Clostridioides*
*difficile* infection (CDI) during treatment.

## 3. Discussion

To the best of our knowledge, this is the first study to compare the clinical effectiveness and adverse effects of SAM and CFZ against BP-MSSA bacteremia. Our results showed no significant differences in the 30- and 90-day mortality rates between the SAM and CFZ groups. Therefore, in cases where CFZ supply is poor or in patients who are allergic to CFZ and PSPs, SAM can be an effective therapeutic option for bacteremia due to BP-MSSA. However, a key point to watch out for is that SAM can cause more adverse events, such as liver dysfunction, than CFZ.

We investigated the efficacy of SAM against BP-MSSA bacteremia because the largest CFZ-producing pharmaceutical factory in Japan experienced manufacturing difficulties in 2019 (the same situation recurred in August 2022). The Ministry of Health, Labor, and Welfare of Japan listed alternative drugs, such as SAM, CRO, and VAN, for use during the CFZ shortage [12]. However, some previous studies have revealed that CRO has a higher rate of treatment failure than CFZ for methicillin-susceptible *Staphylococcus aureus* (MSSA) bacteremia [13,14]. In addition, broad-spectrum antimicrobial agents must be conserved to avoid emerging drug-resistant bacteria (SAM has a narrower spectrum than CRO). Anti-methicillin-resistant *Staphylococcus aureus* agents, such as VAN and DAP, are key drugs for the treatment of MRSA infections. Moreover, VAN is associated with higher rates of infection-related mortality, re-infection, and bacteriologic failure than CFZ or PSPs in the definitive treatment of MSSA bacteremia [15,16]. Therefore, although no study has evaluated the use of SAM in the treatment of BP-MSSA bacteremia, we selected SAM as a definitive therapy against BP-MSSA bacteremia, particularly during the CFZ year of shortage. Our findings suggest that, in addition to preserving broad-spectrum antibiotics, SAM could serve as a comparable alternative to CFZ for the treatment of BP-MSSA bacteremia, as no difference in clinical outcomes was identified. Larger prospective studies are required to confirm our findings.

Bacteremia due to MSSA is associated with high mortality at 20–30% [1,3,17]. Although our study population had relatively low mortality, the 30- and 90-day mortality was 8.5% and 12.7%, respectively. At our hospital, physicians belonging to the antimicrobial stewardship team (AST) evaluate every blood culture-positive patient, recommend empirical therapy with antimicrobial agents, and observe the clinical course carefully to provide important advice as follows: (1) necessary drainage for source control; (2) checking disseminated infections, including endophthalmitis, osteomyelitis, and infective endocarditis; (3) follow blood cultures for assessment of treatment effect; and (4) report adverse effects of selected antibiotics. Moreover, definitive (de-escalation) therapy is recommended as soon as drug susceptibility has been identified. Furthermore, AST physicians follow the clinical course to determine whether a patient’s condition worsens after de-escalation. Although prior studies revealed that the administration of alternative broad-spectrum drugs is associated with worse clinical outcomes [18,19], we could avoid worse deterioration in patient outcomes even in an antimicrobial shortage period. Moreover, based on a multivariate analysis, Uda et al., reported that administration of broad-spectrum antibiotics, such as piperacillin/tazobactam, as definitive therapy was associated with treatment failure in patients with MSSA bacteremia during the CFZ shortage period (odds ratio = 17, *p* = 0.003) [20]. Therefore, careful and restricted intervention via an AST physician is vital during critical antimicrobial shortages, in addition to avoiding prescription of broad-spectrum antibiotics for the treatment of BP-MSSA bacteremia.

In general, CFZ and PSPs are considered first-line drugs for the treatment of MSSA infections [21,22], and both are equally efficacious for treating MSSA bacteremia [7]. Particularly, CFZ has been shown to be more tolerable in the treatment course than PSPs that have high rates of discontinuation owing to adverse events, such as kidney damage. This study showed that SAM treatment for BP-MSSA bacteremia had no deterioration in clinical outcomes when compared with CFZ. Although the investigated number was small, initiating SAM as definitive therapy within 5 days may be an appropriate therapy for BP-MSSA bacteremia, with attention to the adverse effects, particularly liver damage. In addition, almost all antimicrobial agents change the natural flora of the intestines; in particular, ampicillin leads to the overgrowth of *C. difficile* and consequent development of CDI [23]. However, in this study, CDI was not common in SAM compared with CFZ group.

Historically, the inoculum effect (InE) is reported as a phenomenon that affects a significant increase in the minimum inhibitory concentrations (MICs) of antibiotics, particularly in beta-lactamase producing bacteria, such as BP-MSSA [24]. Owing to the InE effect, the MIC increased four-fold higher than the standard inoculum, changing the interpretation from susceptible to resistant, which induced an unfavorable prognosis for MSSA bacteremia [25]. Although previous studies mainly reported InE of CFZ in patients with MSSA infection [26,27,28], Saeki et al., recently reported a significant increase in MICs of not only CFZ but also SAM against BP-MSSA isolated from blood culture (*n* = 38) [29]. In addition, CFZ InE was associated with increased 30-day mortality, particularly in CABSI or an unknown source of bacteremia due to BP-MSSA, compared with patients infected with BP-MSSA strains without CFZ InE [30]. In the present study, all deceased patients treated with SAM or CFZ had CABSI or unknown entry sites for bacteremia (Table 2). Additional investigations are required to determine whether the InE of SAM leads to treatment failure in BP-MSSA bacteremia in clinical practice.

In the present study, there were no differences in all-cause 30- and 90-day mortality rates between the SAM and CFZ groups. However, the proportion of adverse effects related to SAM administration was higher than that with CFZ administration. Furthermore, in general, CFZ (1–2 g, three times daily) is cheaper than SAM (1.5–3 g, four times daily). Therefore, when CFZ is available, we recommend that it should be administered for BP-MSSA bacteremia rather than SAM.

This study has certain limitations. First, it was a retrospective study with a small sample size. However, no studies have compared the clinical effectiveness of SAM and CFZ for BP-MSSA bacteremia. In addition, as experienced in 2019, a problem with a stable supply of antibiotics may occur anywhere. In fact, various countries have recently faced antimicrobial shortages, making first-line treatments unavailable [31]; therefore, our data will help when treating BP-MSSA bacteremia in such situations, in addition to cases of allergy to CFZ or PSPs. Second, we selected patients who received SAM or CFZ as definitive therapy within 5 days of the initial treatment. Thus, we were unable to assess the true effect of SAM or CFZ alone against BP-MSSA bacteremia. The initially selected antimicrobial agents might have affected prognosis, despite no significant differences in selected empirical therapy between the two groups (Table 3). Moreover, in general, approximately 3–4 days are needed to identify the causative pathogen and drug susceptibility, including the results of the zone edge test in clinical practice. Therefore, we recommend definitive therapy for a relatively short time and believe that the results of this study reflect the actual clinical situation from empiric treatment to switching definitive therapy. Fourth, most patients in this study had mild severity (only one patient used vasopressors). Additional studies including patients with severe BP-MSSA bacteremia are needed to evaluate the efficacy of SAM. Fifth, we did not investigate the pathogenicity of the isolated BP-MSSA, such as the virulence genes and sequence type, in addition to phylogenetic analysis. Additional studies including multilocus sequence typing, polymerase chain reaction, and random amplified polymorphic DNAs are also needed to clarify whether these bacterial factors are associated with the outcome of BP-MSSA bacteremia besides selected antimicrobial agents. 

## 4. Materials and Methods

### 4.1. Patient Population

This retrospective cohort study was conducted at Aichi Medical University Hospital (an acute care hospital with approximately 900 inpatient beds) in Aichi Prefecture, Japan. 

Patients older than 18 years who received SAM or CFZ alone as definitive therapy for BP-MSSA bacteremia between April 2014 and March 2022 were screened, and their medical records were reviewed.

### 4.2. Inclusion Criteria

Patients with BP-MSSA bacteremia who received SAM or CFZ for more than 7 days as a definitive therapy within 5 days since initial treatment were included in the study.

### 4.3. Exclusion Criteria

Patients were excluded if they met the following criteria: (1) concurrent use of other antibiotics; (2) did not receive SAM or CFZ as definitive therapy within 5 days of initial antibiotic treatment; and (3) administered SAM or CFZ for less than 7 days as definitive therapy.

### 4.4. Data Collection

Information on the patient’s background (such as age, sex, underlying disease, hospital onset), laboratory results (white blood cell count, C-reactive protein, creatine, albumin), primary site of bacteremia, outcome, initial antimicrobial agent use (empiric therapy), antibiotic-related adverse effects, and CDI complication was retrospectively collected through chart review. The patient’s general condition was evaluated using ECOG PS [32]. ECOG PS is a score ranging from zero to four: zero indicates that the individual is fully active and able to carry on all pre-disease performance without restriction; one indicates some restriction in the performance of physically strenuous activity, but the patient is still ambulatory and able to carry out work of a light or sedentary nature; two indicates the patient is ambulatory and up and about for >50% of waking hours, capable of all self-care, but unable to carry out any work activities; three indicates the patient is capable of limited self-care and confined to a bed or chair for more than 50% of waking hours; and four indicates the patient is completely disabled, unable to carry out any self-care, and totally confined to a bed or chair [33].

The clinical outcomes and antibiotic-related adverse reactions were recorded for each patient. The qSOFA score was used to screen for sepsis [34]. The qSOFA simplifies the SOFA score drastically by including only 3 of its clinical criteria. The qSOFA score includes 1 point for each of the 3 criteria: (1) altered mental status, (2) systolic blood pressure ≤ 100 mm Hg, or (3) respiratory rate ≥ 22 breaths/min. A qSOFA score ≥ 2 is suggestive of sepsis [34]. A previous study regarding SAB revealed that those with initial qSOFA score of ≥2 had a fatal outcome when compared with patients with a qSOFA score < 2 [35]. Disease severity was evaluated using the SOFA score, which is useful in predicting the clinical outcomes of critically ill patients [36,37]. The SOFA score evaluates 6 organs systems (respiratory, coagulation, hepatic, cardiac, neurologic, and renal system), with 0 considered normal and 4 indicating a severe degree of dysfunction [38]. The CCI is calculated from age and comorbidity category-associated weight from 1 to 6 to estimate the mortality risk [39]. A previous study reported that CCI ≥ 4 was an independent factor for 30-day mortality in patients with SAB [40]. Comorbidities of CCI are as follows: myocardial infarction, congestive heart failure, peripheral vascular disease, cerebrovascular disease, dementia, chronic pulmonary disease, collagen disease, peptic ulcer, liver disease (mild or moderate to severe), hemiplegia, chronic kidney disease, diabetes mellitus (uncomplicated or with end-organ damage), malignancy (leukemia, lymphoma, solid tumor, or metastatic solid tumor), and acquired immunodeficiency syndrome.

### 4.5. Hematological and Blood Chemistry Analyses

Hematological analysis was performed using Sysmex XN-9000 automatic hematology analyzer (XN, Sysmex, Kobe, Japan), while blood chemistry was analyzed by the Hitachi LABOSPECT 008AS platform (LABOSPECT, Hitachi High-Tech Co., Tokyo, Japan).

### 4.6. Microbiological Evaluation

Two sets of peripheral blood cultures were collected from the peripheral vein of all patients included in the present study before administering empirical therapy. Isolated *S. aureus* from blood cultures was identified using matrix-assisted laser desorption/ionization time-of-flight mass spectrometry (Bruker Daltonics, Billerica, MA, USA). Drug susceptibility was identified using a “RAISUS RSCES” automated system (Nissui Pharmaceuticals Co., Ltd., Tokyo, Japan). To detect BP-MSSA, the penicillin zone edge test was performed on Mueller–Hinton agar using a 10-U penicillin disk (Eiken Chemical, Tokyo, Japan), according to the Clinical and Laboratory Standards Institute manual [41,42]. Sharp zone edges were interpreted as positive, and all *S. aureus* samples included in this study showed positive results.

### 4.7. Statistical Analysis

Data for categorical variables are expressed as percentages and continuous variables as mean ± standard deviation (SD). The chi-square or Fisher’s exact test (two-tailed) was used to compare categorical variables, and unpaired Student’s *t*-test or the Mann–Whitney U test to compare continuous variables. Statistical analyses were performed using Statistical Package for the Social Sciences version 26 for Windows (SPSS Inc., Chicago, IL, USA). A *p*-value < 0.05 indicated statistical significance. Overall survival was calculated as the time between the date of diagnosis and the date of death as a result of any cause. Kaplan–Meier analysis was conducted using Graph Pad Prism v 9.3.1 (graph Pad, USA). The Generalized Wilcoxon test and log-rank test were used to evaluate significance.

### 4.8. Ethical Considerations

This study was conducted in accordance with the guidelines of the Declaration of Helsinki. This study was approved by the Institutional Review Board of Aichi Medical University Hospital (approval number: 2022-095). The requirement for patient consent was waived owing to the retrospective nature of the study. 

## 5. Conclusions

We investigated the clinical effect of SAM as definitive therapy within 5 days of empirical therapy against BP-MSSA bacteremia and showed that SAM treatment was unassociated with differences in 30- and 90-day mortality when compared with CFZ. To further evaluate the usefulness of SAM for bacteremia due to BP-MSSA, a well-designed prospective study with a large population is needed.

## Figures and Tables

**Figure 1 antibiotics-11-01505-f001:**
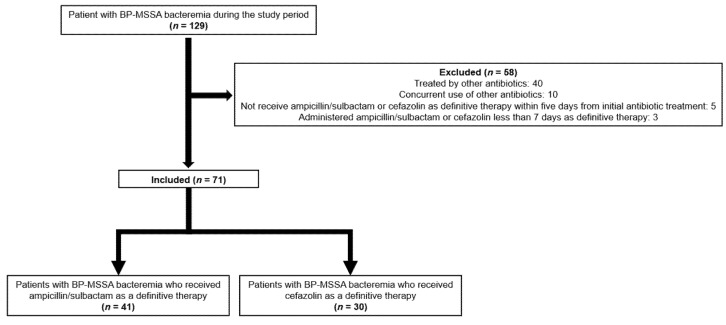
Flowchart of the inclusion and exclusion of β-lactamase-producing methicillin-susceptible *S. aureus* bacteremia cases in the study cohort. Patients were excluded if they met the following conditions: (1) concurrent use of other antibiotics, (2) non-administration of SAM or CFZ as definitive therapy within 5 days from initiation of the initial antibiotic treatment, and (3) administration of SAM or CFZ for less than 7 days as definitive therapy. BP-MSSA; β-lactamase-producing methicillin-sensitive *S. aureus*.

**Figure 2 antibiotics-11-01505-f002:**
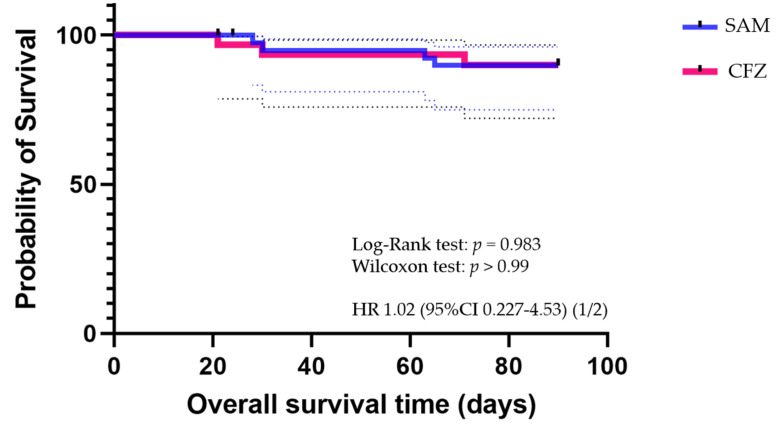
Kaplan–Meier survival curves of the SAM and CFZ cohort. SAM, ampicillin/sulbactam; CFZ, cefazolin; HR, hazard ratio; CI, confidence interval.

**Table 1 antibiotics-11-01505-t001:** Characteristics of the participants.

Variables	All Patients	SAM	CFZ	*p*-Value
	*N* = 71	*N* = 41	*N* = 30	
Mean age (years ± SD)	69.8 ± 18.7	71.3 ± 2.9	67.7 ± 3.4	0.429
Median age (years, range)	75 (18–97)	76 (19–97)	73.5 (18–91)	
Male (*n*, %)	51 (71.8)	29 (70.7)	22 (73.3)	0.809
Body mass index (kg/cm^2^)	25.0 ± 4.8	20.3 ± 0.6	21.6 ± 0.8	0.181
Hospital-acquired bacteremia	31 (43.7)	16 (39.0)	15 (50.0)	0.357
PS (mean ± SD) PS ≥ 3–4 (*n*, %)	1.7 ± 1.4 27 (38.0)	1.9 ± 0.2 18 (43.9)	1.5 ± 0.3 9 (30.0)	0.245 0.233
qSOFA ≥ 2 (*n*, %)	14 (19.7)	9 (22.0)	5 (16.7)	0.580
Septic shock (*n*, %)	7 (9.9)	4 (9.8)	3 (10.0)	0.972
Vasopressor use (*n*, %)	1 (1.4)	1 (2.4)	0 (0)	0.389
Developed disseminated intravascular coagulation (*n*, %)	25 (35.2)	17 (41.5)	8 (26.7)	0.197
Disease severity				
SOFA ≥ 4 (*n*, %)	17 (23.9)	12 (29.2)	5 (16.7)	0.219
Underlying disease				
Atopic dermatitis (*n*, %)	13 (18.3)	8 (19.5)	5 (16.7)	0.759
Diabetes mellitus (*n*, %)	24 (33.8)	14 (34.2)	10 (33.3)	0.943
Hemodialysis (*n*, %)	8 (11.3)	3 (7.3)	5 (16.7)	0.269
Cardiovascular disease (*n*, %)	20 (28.2)	11 (26.8)	9 (30.0)	0.769
Cerebrovascular disease (*n*, %)	20 (28.2)	13 (31.7)	7 (23.3)	0.438
Pulmonary disease (*n*, %)	8 (11.3)	3 (7.3)	5 (16.7)	0.269
Renal dysfunction (*n*, %)	21 (29.6)	13 (31.7)	8 (26.7)	0.645
Solid tumor (*n*, %)	15 (21.1)	6 (14.6)	9 (30)	0.117
Hematological malignancy (*n*, %)	3 (4.2)	1 (2.4)	2 (6.7)	0.569
Autoimmune disease (*n*, %)	3 (4.2)	3 (7.3)	0 (0.0)	0.257
Under chemotherapy (*n*, %)	4 (5.6)	1 (2.4)	3 (10.0)	0.303
Immunosuppressant (*n*, %)	10 (14.1)	6 (14.6)	4 (13.3)	0.876
Charlson Comorbidity Index ≥ 4 (*n*, %)	25 (35.2)	13 (31.7)	12 (40)	0.470
Laboratory data (mean ± SD)				
White blood cell (μL)	13,697 ± 6707	14,356.1 ± 1048.1	12,796.7 ± 1225.3	0.336
C-reactive protein (mg/dL)	16.0 ± 9.2	16.9 ± 1.4	14.7 ± 1.7	0.324
Creatinine (U/L)	1.6 ± 2.2	1.8 ± 0.3	1.4 ± 0.4	0.549
Albumin (g/dL)	2.8 ± 0.6	2.7 ± 0.1	2.9 ± 0.1	0.325
Primary site of infection (*n*, %)				
Primary bacteremia (entry sites were unknown)	25 (35.2)	13 (31.7)	12 (40)	0.469
Secondary bacteremia				
Catheter-associated bloodstream infection	14 (19.7)	7 (17.1)	7 (23.3)	0.512
Skin and soft tissue infection	13 (18.3)	7 (17.1)	6 (20)	0.752
Urinary tract infection	4 (5.6)	2 (4.9)	2 (6.7)	0.746
Infective endocarditis	3 (4.2)	2 (4.9)	1 (3.3)	0.749
Pneumonia	4 (5.6)	4 (9.8)	0 (0.0)	0.132
Cholangitis	1 (1.4)	1 (2.4)	0 (0.0)	1.000
Iliopsoas muscle abscess	2 (2.8)	2 (4.9)	0 (0.0)	0.505
Shunt infection	2 (2.8)	2 (4.9)	0 (0.0)	0.505
Subcutaneous abscess	1 (1.4)	1 (2.4)	0 (0.0)	1.000
Spondylitis	1 (1.4)	0 (0.0)	1 (3.3)	0.422
Surgical site infection	1 (1.4)	0 (0.0)	1 (3.3)	0.422
Metastatic infection	9 (12.7)	6 (14.6)	3 (10)	0.724
Follow-up two sets of blood culture (*n*, %)	54 (76.1)	30 (73.2)	24 (80)	0.505
Continuous positive blood culture for > 7 days (*n*, %)	4 (5.6)	2 (4.8)	2 (6.7)	0.746
Duration of antibiotic treatment (mean ± SD)	29.1 ± 47.4	33.5 ± 6.9	23.5 ± 8.1	0.351
Duration of intravenous treatment (mean ± SD)	17.2 ± 10.7	18.2 ± 1.7	17.0 ± 2.0	0.630
Length of hospital stay of community-acquired MSSA infection (mean ± SD)	36.4 ± 60.9	47.4 ± 10.3	28.3 ± 13.6	0.270
All-cause 30-day mortality (*n*, %)	6 (8.5)	4 (9.7)	2 (6.6)	0.643

SAM, ampicillin/sulbactam; CFZ, cefazolin; SD, standard deviation; qSOFA, the Quick Sepsis-related Organ Failure Assessment; SOFA, Sepsis-related Organ Failure Assessment; PBS, Pitt Bacteremia Score; PS, performance status; MSSA, methicillin-susceptible *S. aureus*.

**Table 2 antibiotics-11-01505-t002:** Initially administered antibiotics for beta-lactamase-producing MSSA in this study population.

Initial Antimicrobial Agents	All Patients (*n*, %)	SAM (*n*, %)	CFZ (*n*, %)	*p*-Value
	*N* = 71	*N* = 41	*N* = 30	
Ampicillin/sulbactam	14 (19.7)	11 (26.8)	3 (10.0)	0.129
Piperacillin/tazobactam	12 (17.0)	7 (17.1)	5 (16.7)	0.964
Cefazolin	5 (7.0)	0 (0.0)	5 (16.7)	0.010
Cefmetazole	1 (1.4)	1 (2.4)	0 (0.0)	1.000
Ceftriaxone	11 (15.5)	7 (17.1)	4 (13.3)	0.749
Cefozopran/sulbactam	1 (1.4)	1 (2.4)	0 (0.0)	1.000
Cefepime	1 (1.4)	1 (2.4)	0 (0.0)	1.000
Doripenem	1 (1.4)	1 (2.4)	0 (0.0)	1.000
Meropenem	8 (11.3)	3 (7.3)	5 (16.7)	0.269
Vancomycin	8 (11.3)	5 (12.2)	3 (10.0)	1.000
Teicoplanin	4 (5.6)	3 (7.3)	1 (3.3)	0.632
Daptomycin	4 (5.6)	1 (2.4)	3 (10.0)	0.303
Levofloxacin	1 (1.4)	0 (0.0)	1 (3.3)	0.422

SAM, ampicillin/sulbactam; CFZ, cefazolin; MSSA, methicillin-susceptible *S. aureus.*

**Table 3 antibiotics-11-01505-t003:** Associated side effects of SAM or CFZ.

Variables	All Patients	SAM	CFZ	*p*-Value
	*N* = 71	*N* = 41	*N* = 30	
None (*n*, %)	49 (69.0)	24 (58.5)	25 (83.3)	0.026
Diarrhea (*n*, %)	7 (9.8)	6 (14.6)	1 (3.3)	0.115
Elevated eosinocyte levels (*n*, %)	4 (5.6)	2 (4.8)	2 (6.6)	0.747
Drug rush (*n*, %)	4 (5.6)	3 (7.3)	1 (3.3)	0.472
Liver dysfunction (*n*, %)	13 (18.3)	11 (26.8)	2 (6.6)	0.030
Thrombopenia (*n*, %)	1 (1.4)	0	1 (3.3)	0.239
Complicated CDI	0	0	0	-

SAM, ampicillin/sulbactam; CFZ, cefazolin; CDI, *Clostridium difficile* infection.

## Data Availability

Data sharing is not applicable to the present study.

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
