# Peer review of "Comparative Effectiveness of Ampicillin/Sulbactam versus Cefazolin as Targeted Therapy for Bacteremia Caused by Beta-Lactamase-Producing Methicillin-Sensitive Staphylococcus aureus: A Single-Center Retrospective Study"

_antibiotics, 2022, doi:10.3390/antibiotics11111505_

Round 1
Reviewer 1 Report
This retrospective study observed Ampicillin/sulbactam (SAM) as an alternative agent for beta-lactamase-producing methicillin-susceptible Staphylococcus aureus (BP-MSSA) infection and compared with CFZ.
The introduction is in well written. A CFZ shortage is encountered in Japan and this text could bridge a knowledge gap for unavailable CFZ in first-line treatments.
In your method section, Charlson Comorbidity Index was applied to measure. But the detail about this index was not clear. If possible, this section is suggested to present. Also, about other score such as Quick Sepsis-related Organ Failure Assessment (qSOFA) score, items composition should be listed in your text for readers to refer. Other scores such as SOFA score et al. are recommended to present in text.
In results section, “None of the patients had complications of Clostridium difficile infection (CDI) during treatment” was mentioned. Why Clostridium difficile infection (CDI) was paid attention on?
Reviewer 2 Report
The research can be improved after conducting Multilocus sequence typing (MLST) and PCR for detection of virulence genes and pathogenicity ilsands of the Staphylococcus aureus isolates. Also, RAPD or RFLP can be used for phylogenetic analysis.
According to the obtained results, the authors have to expand the discussion.
Reviewer 3 Report
The manuscript by Jun Hirai et al. describes the clinical differences of patients under treatment with SAM and Cefazolin for bacteremia caused by methicillin-susceptible S. aureus.
The manuscript needs to be extensively reprinted to be understood.
The title should be changed to reflect that these are methicillin-sensitive strains in addition to beta-lactamase positive.
A microbiologist should be added to the list of authors, given the abundance of bacteriological results.
Italicize "vs."; "et al."
How the authors considered the correction for multiplicity of statistical tests, it is for example unlikely that PBS or CCI>5 are really significantly different after correction.
Line 105. Attention absence of evidence is not evidence of absence of difference.
Line 123: diarrhoea is not a significant parameter according to table 3.
Line 126: Clostridium must be replaced by Clostridioides.
The major bias is that the manuscript concludes to equivalence of the two antibiotics management but is designed as a superiority assay. The number of patients to include must be calculated accordingly and justified.
Prefer passive voice.
Inclusion criteria (paragraph 4.3) must be placed before exclusion criteria (paragraph 4.2.)
Methodologies/assays for determination of WBC, CRP, albumin and creatinine must be appropriately referenced.
Round 2
Reviewer 2 Report
The authors have responded to the reviewers' recommendations and questions.
Reviewer 3 Report
To my opinion, the manuscript has be well-improved, according to my previous comments and now, could be published.